# Source Identification of a Chemical Incident in an Urban Area

Francisco J. Fernández [1,*,†] and Miguel E. Vázquez-Méndez [2,†]

1   Instituto de Matemáticas, Universidade de Santiago de Compostela, 15782 Santiago de Compostela, Spain
2   Departamento Matemática Aplicada, Universidade de Santiago de Compostela, EPSE, 27002 Lugo, Spain; miguelernesto.vazquez@usc.es
*   Correspondence: fjavier.fernandez@usc.es
†   These authors contributed equally to this work.

**Abstract:** This work deals aims to present a methodology for source identification of chemical incidents in urban areas. We propose an approximation of the problem within the framework of the optimal control theory and we provide an algorithm for its numerical resolution. Finally, we analyze the validity of the algorithm in several academic situations.

**Keywords:** chemical incident; source identification; control problem; mixed-integer nonlinear programming; numerical simulation

## 1. Introduction

The existence of weapons of mass destruction represents an increase in the potential threat to peace in different areas of the world. Some of these weapons, with capacity for killing and bringing significant harm to numerous humans, may not only be in the hands of great powers, but also of regional powers and even terrorist organizations.

Regarding the use of nuclear, biological, chemical (NBC) weapons, this possibility must be always considered in asymmetric armed conflicts, where their use is more likely than in conflicts between great powers. An adversary with NBC capacity may introduce agents, materials and weapons at any time in a more or less indirect way. The launch and dispersal of NBC agents or materials can be carried out with different means such as missiles, aircrafts of all kinds, field artillery, difficult-to-detect aerosols, etc. Once an NBC incident occurs, it is vitally important to be able to predict the danger area to evacuate the civilian population and alert the mobilized units in that area [1]. The methods used to estimate the hazard area can be classified into:

(i)   Simplified methods: These are preliminary estimates based on the characteristics of the incident and meteorological data. They are usually carried out by hand by a trained person.
(ii)  Improved methods: These are automatic or manual estimates that are usually made taking into account the type of incident, the place where it occurred, and the weather conditions. They are more accurate than previous ones and update as weather conditions change.
(iii) Methods based on mathematical simulation: These are fully automatic methods that estimate the hazard area by numerical simulation, from the type of incident, meteorological data, and space-time domain information.

Currently, the most widely used mathematical models to simulate the evolution of a chemical agent are based on the Gaussian models. They are closed-form analytical solutions of the classic advection-diffusion equation

$$\frac{\partial c}{\partial t} + \nabla \cdot (c\mathbf{u}) = \nabla \cdot (\mathbf{K}\nabla c) + S, \quad \Omega \times ]0, T[, \tag{1}$$

derived from some suitable hypotheses [2,3]. Specifically, it is assumed that:

**Hypothesis 1.** *The concentration of the chemical agent c is not dependent on time (the solution is steady state) and, specifically, $\partial c / \partial t = 0$.*

**Hypothesis 2.** *The incident occurs at a fixed point $\mathbf{b} = (0, 0, H)$, where the chemical agent is emitted at a constant rate $Q > 0$, in such a way that the source term is given by $S(\mathbf{x}) = Q\delta_{\mathbf{b}}(\mathbf{x})$, where $\delta_{\mathbf{b}}(\mathbf{x})$ is the Dirac delta at point $\mathbf{b}$.*

**Hypothesis 3.** *The wind velocity field $\mathbf{u}$ is constant and aligned with the positive $x_1$-axis, that is $\mathbf{u} = (u, 0, 0)$ for some constant $u \geq 0$.*

**Hypothesis 4.** *The diffusion coefficients are the same in all directions and only depend on the downwind distance $x_1$, that is, the diffusion matrix $\mathbf{K}$ is given by $\mathbf{K} = K(x_1)I_3$, where $K(x_1)$ is a real function and $I_3$ is the identity matrix.*

**Hypothesis 5.** *The effect of diffusion on the $x_1$-axis is neglected (dominant convection), that is, $K(x_1)\partial^2 c / \partial x_1{}^2 = 0$.*

**Hypothesis 6.** *Topographic variations and obstacles (trees, buildings, etc.) are neglected, in such a way that the space domain is $\Omega = [0, \infty) \times (-\infty, \infty) \times [0, \infty)$.*

**Hypothesis 7.** *There is no chemical agent for $x_1 < 0$, and the chemical agent does not penetrate the soil. Thus, the following boundary conditions can be considered*

$$
\begin{aligned}
&c(0, x_2, x_3) = c(+\infty, x_2, x_3) = 0, \ \forall x_2 \in \mathbb{R}, x_3 \geq 0, \\
&c(x_1, -\infty, x_3) = c(x_1, +\infty, x_3) = 0, \ \forall x_1, \ x_3 \geq 0, \\
&K(x_1)\partial c / \partial x_3(x_1, x_2, 0) = 0, \ c(x_1, x_2, +\infty) = 0, \ \forall x_1 \geq 0, \ x_2 \in \mathbb{R}.
\end{aligned}
\tag{2}
$$

Under these hypotheses, the general Equation (1) is rewritten as

$$
u \frac{\partial c}{\partial x_1} = K \left( \frac{\partial^2 c}{\partial x_2{}^2} + \frac{\partial^2 c}{\partial x_3{}^2} \right) + Q\delta_{\mathbf{b}}(\mathbf{x}), \quad \Omega
\tag{3}
$$

and completed with condition (2). The explicit solution of systems (2) and (3) is obtained by using the Laplace transform [3], and it is known as the *Gaussian-plume* model:

$$
c(r, x_2, x_3) = \frac{Q\exp\left(-\frac{(x_2)^2}{4r}\right)\left(\exp\left(-\frac{(x_3-H)^2}{4r}\right) + \exp\left(-\frac{(x_3+H)^2}{4r}\right)\right)}{4\pi u r},
\tag{4}
$$

where

$$
r = \frac{1}{u} \int_0^{x_1} K(\xi)\, d\xi.
$$

If the chemical incident is instantaneous and only occurs at the initial time $t = 0$, the Gaussian-plume model is not suitable. In this situation, hypotheses (H$_3$)–(H$_7$) hold, but (H$_1$) and (H$_2$) must be replaced respectively by:

**Hypothesis 8.** *There is no chemical agent before the incident, that is,*

$$
c(x_1, x_2, x_3, 0) = 0, \quad \forall x_1, x_3 \geq 0, x_2 \in \mathbb{R}.
\tag{5}
$$

**Hypothesis 9.** *The incident occurs at the initial time $t = 0$, and at a fixed point $\mathbf{b} = (0, 0, H)$, in such a way that the source term is given by $S(\mathbf{x}) = Q_T \delta_{\mathbf{b}}(\mathbf{x})\delta_0(t)$, where $Q_T$ is the total amount of chemical agent released.*

Under these hypotheses, the general Equation (1) is rewritten as

$$\frac{\partial c}{\partial t} + u\frac{\partial c}{\partial x_1} = K\left(\frac{\partial^2 c}{\partial x_2{}^2} + \frac{\partial^2 c}{\partial x_3{}^2}\right) + Q_T \delta_{\mathbf{b}}(\mathbf{x})\delta_0(t), \quad \Omega \times ]0, T[,$$

and completed with the initial condition (5), and boundary conditions (2) formulated as $\forall t > 0$. The system is solved again by using the Laplace transform, and the explicit solution

$$c(r, x_2, x_3, t) = \frac{Q_T \exp\left(-\frac{(x_1 - ut)^2 + (x_2)^2}{4r}\right)\left(\exp\left(-\frac{(x_3 - H)^2}{4r}\right) + \exp\left(-\frac{(x_3 + H)^2}{4r}\right)\right)}{8(\pi r)^{3/2}} \quad (6)$$

is referred as the *Gaussian-puff* model.

Previous considerations (hypotheses (H$_3$)–(H$_7$)) clearly reveal the limitations of Gaussian models (4) and (6) to simulate the evolution of a chemical agent in an urban region, and consequently, to determine the hazard area if the chemical incident occurs in an urban domain. The main objective of this paper is just to develop a novel method to deal with chemical incidents in urban areas. To avoid the limitations of the Gaussian models, we will deal with the general equation, Equation (1), to characterize the source of the chemical agent, from measurements made at atmospheric monitoring stations located at different points of the city. The scientific literature on this subject is very rich, and there are many papers dealing not only with the mathematical study of inverse source problems [4–6], but also with interesting environmental applications in surface water [7–10], in groundwater [11,12] and in the atmosphere [2,8]. In this paper, the problem will be studied within the framework of optimal control problem of partial differential equations (PDEs). Taking advantage of previous works of the authors on the control of the urban heat island [13,14], and thinking about a 3D urban domain, the main novelty of the model proposed in this paper is that the classic advection-diffusion equation, Equation (1), will be completed with a reaction term depending on the air temperature, and combined with a 3D microclimatic model to simulate the wind velocity between buildings and the heat transfer between air, soil and buildings. Additionally, taking into account that the admissible set may be nonconnected, the inverse problems will be formulated and solved within the framework of mixed integer nonlinear programming (MINLP).

This paper is organized as follows. The mathematical model proposed to simulate the chemical agent evolution is presented in Section 2.1, and completed in Appendix A, where the 3D microclimatic model is detailed. From this model, in Section 2.2 MINLP is used to formulate the inverse problems within the framework of optimal control of PDEs. A completed numerical method to solve these problems is detailed in Section 2.3, and numerical results are presented and discussed in Section 3. Finally, some conclusions are summarized in Section 4.

## 2. Materials and Methods

### 2.1. Numerical Simulation: The State Model

In this section, we present the 3D mathematical model that we will use in the numerical resolution of the problem. We will consider a three-dimensional bounded domain $\Omega_A^{3D} \subset \mathbb{R}^3$ corresponding to an urban area, where $\partial\Omega_A^{3D}$ is the boundary of said domain, this is walls and ceilings of buildings, floor and, additionally, fictitious borders that delimit our domain (see Figure 1). We will denote by $\Gamma_A^{IN}$ the boundary corresponding to an incoming air flow. We will assume that the air temperature can affect the concentration of the chemical agent and that, eventually, there may be sedimentation effects. Therefore, the evolution of the concentration of the chemical agent $c_A$ (gr/m$^3$) will be given by the solution of the following equation:

$$\begin{cases} \dfrac{\partial c_A}{\partial t} + \mathbf{u}_A \cdot \nabla c_A + w_C \dfrac{\partial c_A}{\partial x_3} - \nabla \cdot (K_C \nabla c_A) = F_C + G(\theta_A, c_A), \quad \Omega_A^{3D} \times ]0, T[, \\[2mm] c_A = c_A^{IN}, \quad \Gamma_A^{IN} \times ]0, T[, \\[2mm] K_C \dfrac{\partial c_A}{\partial \mathbf{n}_A} = 0, \quad (\partial \Omega_A^{3D} \setminus \Gamma_A^{IN}) \times ]0, T[, \\[2mm] c_A(0) = c_A^0, \quad \Omega_A^{3D}. \end{cases} \tag{7}$$

where $w_C$ (m/s) is the sedimentation velocity (constant), $\mathbf{u}_A$ (m/s) is the air velocity, $\theta_A$ (K) is the air temperature, $F_C$ (gr/m$^3$ s) is the source term, $G$ (gr/m$^3$ s) represents the influence of air temperature on the chemical agent (in order to simplify the model we will assume that $G(\theta_A, c_A) = G(\theta_A) c_A$, with $G(\cdot)$ a negative function), $K_A$ (m$^2$/s) is the diffusion constant, $c_A^{IN}$ (gr/m$^3$) is the concentration of the chemical agent on the inlet boundary $\Gamma_A^{IN}$, and $c_A^0$ (gr/m$^3$) is the initial concentration of the chemical agent. We must mention that $\nabla \cdot (\mathbf{u_A} c) = \nabla \cdot \mathbf{u}_A c_A + \mathbf{u}_A \cdot \nabla c_A = \mathbf{u} \cdot \nabla c_A$ since we are assuming that $\nabla \cdot \mathbf{u}_A = 0$ (when we are considering air layers close to the ground, it is usually considered that the air behaves like an incompressible fluid). Regarding the source term, it is frequently considered [15] to be of the form:

$$F_C(\mathbf{x}, t) = Q_C(t) \delta_{\mathbf{b}_C}(\mathbf{x}), \tag{8}$$

where $Q_C(t)$ (gr/s) is the release rate and $\mathbf{b}_C$ is the point in which the source term is located. In the case of an instantaneous release, we will consider the following term:

$$F_C(\mathbf{x}, t) = Q_C \delta_0(t) \delta_{\mathbf{b}_C}(\mathbf{x}), \tag{9}$$

where $Q_C$ (gr) is the total amount of chemical agent released at time $t = 0$. In the case we have an instantaneous release, we will rewrite (9) in terms of an initial condition:

$$c_A(0) = Q_C \delta_{\mathbf{b}_C}(\mathbf{x}).$$

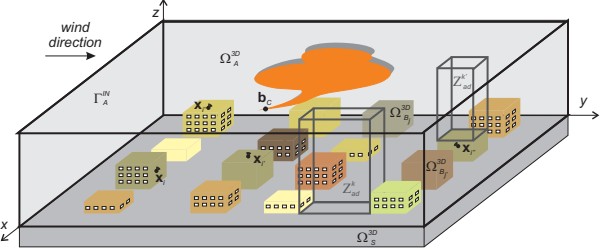

**Figure 1.** Scheme of a 3D urban domain where a chemical incident occurs at point $\mathbf{b}_C$.

The temperature $\theta_A$ (K) and air velocity $\mathbf{u}_A$ (m/s) will be obtained by solving a microclimatic model in which we will take into account the temperature of the soil $\theta_S$ (K) and buildings $\theta_B$ [K] (see Appendix A).

*2.2. Optimal Control: The Inverse Problem*

In this section, we will formulate the inverse problem consisting of the characterization of the source term associated with the chemical incident from a set of measurements taken in the urban area. For this, we will formulate the inverse problem by means of an optimal control problem [16]. Let us start by establishing the following notations:

- We will assume that the possible release point locations ($\mathbf{b}_C$) are in a bounded region $\mathcal{U}_{ad}$ given by the union of $M$ convex closed and bounded subsets (admissible release zones):

$$\mathcal{U}_{ad} = \bigcup_{k=1}^{M} Z_{ad}^k,$$

where $\text{int}(Z_{ad}^i) \cap \text{int}(Z_{ad}^j) = \emptyset, \forall i \neq j$, and

$$Z_{ad}^k = [l_1^k, u_1^k] \times [l_2^k, u_2^k] \times [l_3^k, u_3^k], \quad k = 1, \ldots, M,$$

where $l_i^k$ and $u_i^k$ are, respectively, the lower and upper bounds for the $x_i$ coordinate of $Z_{ad}^k$, $i = 1, 2, 3$. Let us empathize that the set $\mathcal{U}_{ad}$ can be nonconnected.
- $Q_C$ is the release rate or the total amount of chemical agent released. We will asume that $Q_C \in \mathcal{V}_{ad}$, where $\mathcal{V}_{ad} = \{Q \in X : 0 \leq Q(t) \leq Q_{max}, \forall t \in [0, T]\}$ if the source term is given by (8) or $\mathcal{V}_{ad} = \{Q \in X : 0 \leq Q_C \leq Q_{max}\}$ if the source term is given by (9), where $X = L^2(0, T)$ in the first case and $X = \mathbb{R}$ in the second one.
- $\{\widetilde{c}_i^n : i = 1, \ldots, N_P, n = 1, \ldots, N_T\}$ is the set of measurements taken in the urban area at points $(\mathbf{x}_i, t_n) \in (\overline{\Omega_A^{3D}} \setminus \Gamma_A^{IN}) \times [0, T]$, $i = 1, \ldots, N_P$, $n = 1, \ldots, N_T$.
- We consider the following objective function:

$$J(\mathbf{b}_C, Q_C) = \sum_{i=1}^{N_P} \sum_{n=1}^{N_T} (c_A(\mathbf{x}_i, t_n) - \widetilde{c}_i^n)^2.$$

We must observe that to evaluate the previous function at each element $(\mathbf{b}_C, Q_C)$ it is necessary to solve the state equation, Equation (7).

We will study the following optimal control problems [16].

- **Problem 1. Estimation of the release point:** We will estimate the release point assuming that the emission rate $\widetilde{Q}_C$ of the chemical agent is known:

$$\min_{\mathbf{b}_C \in \mathcal{U}_{ad}} J(\mathbf{b}_C, \widetilde{Q}_C). \tag{10}$$

- **Problem 2. Estimation of the release rate:** We will estimate the release rate (or the total amount) $Q_C$ assuming that the release point $\widetilde{\mathbf{b}}_C$ is known:

$$\min_{Q_C \in \mathcal{V}_{ad}} J(\widetilde{\mathbf{b}}_C, Q_C). \tag{11}$$

- **Problem 3. Estimation of the release point and rate:**

$$\min_{(\mathbf{b}_C, Q_C) \in \mathcal{U}_{ad} \times \mathcal{V}_{ad}} J(\mathbf{b}_C, Q_C). \tag{12}$$

Let us observe that problems 1 and 3, Equations (10) and (12), respectively, can be formulated as Nonlinear Mixed Integer Programming Problems (MINLPs); if we introduce an integer variable $\mathbf{y}_C \in \{0, 1\}^M$ such that $y_C^k = 1$, if the release point is in the zone $Z_{ad}^k$, and $y_C^k = 0$ in other cases. Taking into account the previous variable, we can reformulate problem 3, Equation (12), in the following classical framework of MINLPs (problem 1, Equation (10), is analogous):

$$\begin{aligned} \min_{(\mathbf{b}_C, Q_C, \mathbf{y}_C) \in \mathbb{R}^3 \times \mathcal{V}_{ad} \times \{0,1\}^M} \quad & J(\mathbf{b}_C, Q_C) \\ \text{s.t.} \quad \mathbf{h}(\mathbf{b}_C, \mathbf{y}_C) \quad &\leq \quad \mathbf{0} \\ A\mathbf{y}_C \quad &= \quad \mathbf{c}, \end{aligned} \tag{13}$$

where:

- $A \in \mathcal{M}_{m_2 \times M}$ with $m_2 = 1$ is such that $A\mathbf{y}_C = \sum_{k=1}^{M} y_C^k$, therefore:

$$A = \begin{pmatrix} 1 & 1 & \cdots & 1 \end{pmatrix},$$

and $\mathbf{c} = 1 \in \mathbb{R}^{m_2}$. Observe that the constraint $A\mathbf{y}_C = \mathbf{c}$ implies that there is only one 1 in the control vector $\mathbf{y}_C$ with the rest of its components being equal to 0. We will denote by

$$\mathcal{Y}_{ad} = \{\mathbf{y}_C \in \{0,1\}^M : A\mathbf{y}_C = \mathbf{c}\}$$

the admissible control set for the discrete variable $\mathbf{y}_C$.

- Given $\mathbf{b}_C = (p_C^1, p_C^2, p_C^3) \in \mathbb{R}^3$ and $\mathbf{y}_C \in \mathcal{Y}_{ad}$,

$$\mathbf{h}(\mathbf{b}_C, \mathbf{y}_C) = \begin{pmatrix} \mathbf{h}^1(\mathbf{b}_C, \mathbf{y}_C) \\ \mathbf{h}^2(\mathbf{b}_C, \mathbf{y}_C) \\ \mathbf{h}^3(\mathbf{b}_C, \mathbf{y}_C) \end{pmatrix} \in \mathbb{R}^6,$$

being

$$\mathbf{h}^j(\mathbf{b}_C, \mathbf{y}_C) = \begin{pmatrix} \sum_{k=1}^{M} y_C^k (l_j^k - p_C^j) \\ \sum_{k=1}^{M} y_C^k (p_C^j - u_j^k) \end{pmatrix} \in \mathbb{R}^2, \; j = 1, 2, 3.$$

Observe that if $\mathbf{y}_C \in \mathcal{Y}_{ad}$ and $\mathbf{b}_C \in \mathbb{R}^3$ satisfies $\mathbf{h}(\mathbf{b}_C, \mathbf{y}_C) \leq 0$, then $\mathbf{b}_C \in \mathcal{U}_{ad}$.

If we fix the variable $\mathbf{y}_C^* \in \mathcal{Y}_{ad}$ we obtain a classical NLP problem:

$$\min_{(\mathbf{b}_C, Q_C) \in \mathbb{R}^3 \times \mathcal{V}_{ad}} \quad J(\mathbf{b}_C, Q_C) \qquad\qquad (14)$$
$$\text{s.t.} \quad \mathbf{h}(\mathbf{b}_C, \mathbf{y}_C^*) \;\leq\; \mathbf{0}.$$

We denote by $(\mathbf{b}_C^*, Q_C^*)$ a solution of the optimal control problem (14) associated with the discrete variable $\mathbf{y}_C^*$. Thus, if we consider the following set:

$$\mathcal{F}(\mathcal{Y}_{ad}) = \{(\mathbf{b}_C^*, Q_C^*, \mathbf{y}_C^*) \in \mathcal{U}_{ad} \times \mathcal{V}_{ad} \times \mathcal{Y}_{ad} :$$
$$(\mathbf{b}_C^*, \mathbf{Q}_C^*) \text{ solution of (14) associated with } \mathbf{y}_C^*\},$$

we can solve problem (13) by taking $(\mathbf{b}_C, \mathbf{Q}_C, \mathbf{y}_C) \in \mathcal{F}(\mathcal{Y}_{ad})$ such that:

$$J(\mathbf{b}_C, \mathbf{Q}_C) = \min\{J(\mathbf{b}_C^*, \mathbf{Q}_C^*) : (\mathbf{b}_C^*, \mathbf{Q}_C^*, \mathbf{y}_C^*) \in \mathcal{F}(\mathcal{Y}_{ad})\}. \qquad (15)$$

For low values of $M$, the size of the set $F(Y_{ad})$ is small, and the MINLP problem, Equation (15), can be easily solved by an exhaustive search. For large-size problems, more appropriate methods, such as Branch and Bound, Generalized Benders Decomposition or external approximation (see, for instance, [17–19]) must be used. In any case, a quick method for solving the NLP problem, Equation (14), is the key for solving the MINLP problem, Equation (15). Consequently, the next section is devoted to present a numerical method for solving the NLP problem, Equation (14).

### 2.3. Numerical Resolution

To solve the NLP problem, Equation (14), we will use an algorithm of interior points, more specifically, IPOPT [20]. The use of this class of interior point algorithm requires, at least, the evaluation of the cost functional and the constraints, the evaluation of the gradient of the cost functional and the Jacobian matrix associated with the constraints. Since the constraints are linear, the problem lies in calculating the cost function and its gradient. In this section, we will detail how to carry out these computations.

The first step is the numerical resolution of the state equation, Equation (7). In order to achieve this, we will propose a space-time discretization based on the method of the characteristics and the finite element method [21,22]. Spatial and time discretizations have been performed in the scientific software FreeFem ++ [23]. For simplicity in the notations, and without loss of generality, we will assume that that the sedimentation rate is incorporated in the term $\mathbf{u}_A \cdot \nabla c_A$ and we will use the notation $\mathbf{u}$ instead of $\mathbf{u}_A$.

Let us consider $N + 1$ points $\{t^n\}_{n=0}^N$ in the interval $[0, T]$ such that:

- $t^0 = 0$,
- $t^N = T$,
- $t^{n+1} - t^n = \Delta t, \quad \forall n = 0, \ldots, N - 1$.

We define $\alpha = \dfrac{1}{\Delta t}$ and we consider the material derivative for a scalar field $c$:

$$\frac{Dc}{Dt}(\mathbf{x}, t) = \frac{\partial}{\partial t} c(X(\mathbf{x}, t), t) = \frac{\partial c}{\partial t}(\mathbf{x}, t) + \mathbf{u}(\mathbf{x}, t) \cdot \nabla c(\mathbf{x}, t),$$

where $\dfrac{\partial X}{\partial t}(\mathbf{x}, t) = \mathbf{u}(\mathbf{x}, t)$. We can consider the following approximation for the material derivative in a time $t^{n+1}$:

$$\frac{Dc}{Dt}(t^{n+1}) \simeq \alpha(c^{n+1} - c^n \circ X_h^n),$$

with $X_h^n(\mathbf{x}) = X(\mathbf{x}, t_{n+1}, t_n)$ being the solution of the following initial value problem:

$$\begin{cases} \dfrac{dX}{d\tau} = \mathbf{u}(X(\mathbf{x}, t, \tau), \tau). \\ X(\mathbf{x}, t, t) = \mathbf{x}. \end{cases}$$

Thus, $c^n \circ X_h^n \simeq c^n(\mathbf{x} - \mathbf{u}_n(\mathbf{x})\Delta t)$. This approximation, as we will see later, is very important for obtaining the gradient of the objective function.

So, given $c_A^0$, we compute $\{c_A^{n+1}\}_{n=0}^{N-1}$ solving the following equation:

$$\begin{cases} \alpha c_A^{n+1} - \nabla \cdot (K_C \nabla c_A^{n+1}) = F_C^{n+1} + G(\theta_A^{n+1})c_A^{n+1} + \alpha(c_A^n \circ X_h^n), & \Omega_A^{3D}, \\ c_A^{n+1} = 0, & \Gamma_A^{IN}, \\ K_C \dfrac{\partial c_A^{n+1}}{\partial \mathbf{n}_A} = 0, & \partial\Omega_A^{3D} \setminus \Gamma_A^{IN}. \end{cases} \tag{16}$$

For spatial discretization, we will assume that the domain $\Omega_A^{3D}$ is polyhedral and we consider $\{\tau_h^A\}_{h>0}$ as a family of regular meshes of the domain $\Omega_A^{3D}$. We define the following finite element space:

$$X_h^A = \{z \in \mathcal{C}(\overline{\Omega_A^{3D}}) : z_{|\mathcal{T}} \in \mathcal{P}_1(\mathcal{T}), \forall \mathcal{T} \in \tau_h^A, z_{|\Gamma_A^{IN}} = 0\}.$$

Thus, the fully discretized problem consists of $\{c_A^{n+1}\}_{n=0}^{N-1} \subset X_h^A$ solving the following variational formulation:

$$\begin{aligned} \alpha \int_{\Omega_A^{3D}} c_A^{n+1} z \, d\mathbf{x} + \int_{\Omega_A^{3D}} K_C \nabla c_A^{n+1} \cdot \nabla z \, d\mathbf{x} - \int_{\Omega_A^{3D}} G(\theta_A^{n+1}) c_A^{n+1} z \, d\mathbf{x} \\ = \int_{\Omega_A^{3D}} F_C^{n+1} z \, d\mathbf{x} + \alpha \int_{\Omega_A^{3D}} (c_A^n \circ X_h^n) z \, d\mathbf{x}, \quad \forall z \in W_h^A. \end{aligned} \tag{17}$$

**Remark 1.** *Taking into account the definition of the Dirac Delta as a distribution, the first addition in the second term of Equation* (17) *can be computed using the following formula:*

$$\int_{\Omega_A^{3D}} F_C^{n+1} z \, d\mathbf{x} = \int_{\Omega} Q_C(t^{n+1}) \delta_{\mathbf{b}_C}(\mathbf{x}) z \, d\mathbf{x} = Q_C(t^{n+1}) z(\mathbf{b}_C). \tag{18}$$

*However, if we want to model a case in which the emission is not strictly punctual, the following approximation of the Dirac Delta can be considered:*

$$\rho(\mathbf{x}) = \begin{cases} C \exp\left(\frac{1}{\|\mathbf{x}\|^2 - 1}\right) & \|x\| < 1 \\ 0 & \|x\| \geq 1 \end{cases}$$

*where* $C >$ *is such that:*

$$\int_{\mathbb{R}^n} \rho(\mathbf{x}) dx = 1.$$

*From the previous function, we can define*

$$\varphi_{\mathbf{q},\epsilon}(\mathbf{x}) = \frac{1}{\epsilon^n} \rho\left(\frac{\mathbf{x} - \mathbf{q}}{\epsilon}\right).$$

*which converges to* $\delta_{\mathbf{q}}(\mathbf{x}) = \delta(\mathbf{x} - \mathbf{q})$ *when* $\epsilon \to 0$. *Taking into account the previous sequence,*

$$F_C(\mathbf{x}, t) \simeq Q_C(t) \varphi_{\mathbf{b}_C, \epsilon}(\mathbf{x}),$$

*thus*

$$\int_{\Omega_A^{3D}} F_C^{n+1} z \, d\mathbf{x} \simeq Q_C^{n+1} \int_{\Omega_A^{3D}} \varphi_{\mathbf{b}_C, \epsilon}(\mathbf{x}) z \, dx. \tag{19}$$

If $F_C$ is given by (8), let us consider the following discretization for the control variable $Q_C(t)$:

$$Q_C(t) = Q_C^1 \chi_{[t_0, t_1]}(t) + \sum_{n=2}^{N} Q_C^n \chi_{(t_{n-1}, t_n]}(t),$$

where $\mathbf{Q}_C = (Q_C^1, Q_C^2, \ldots, Q_C^n) \in \mathbb{R}^n$. Thus,

$$\mathbf{s}_C = (\underbrace{Q_C^1, Q_C^2, \ldots, Q_C^n}_{\mathbf{Q}_C}, \underbrace{p_C^1, p_C^2, p_C^3}_{\mathbf{b}_C}, \underbrace{y_C^1, y_C^2, \ldots, y_C^M}_{\mathbf{y}_C}) \in \mathbb{R}^n \times \mathbb{R}^3 \times \{0, 1\}^M$$

denotes the discrete global control.

On the contrary, if $F_C$ is given by (9), we obtain

$$\mathbf{s}_C = (\underbrace{Q_C}_{\mathbf{Q}_C}, \underbrace{p_C^1, p_C^2, p_C^3}_{\mathbf{b}_C}, \underbrace{y_C^1, y_C^2, \ldots, y_C^M}_{\mathbf{y}_C}) \in \mathbb{R}^n \times \mathbb{R}^3 \times \{0, 1\}^M$$

as the global control variable.

In order to simplify the notations, we will assume that the measurements are made at the times associated with the time discretization. Thus,

$$J(\mathbf{b}_C, \mathbf{Q}_C) = \sum_{i=1}^{N_P} \sum_{n=1}^{N} (c_A^n(\mathbf{x}_i) - \tilde{c}_i^n)^2,$$

is the discretized objective function, where $\{c_A^n\}_{n=1}^N$ are the solutions of the fully discretized state equation, Equation (17).

To calculate the gradient of the cost functional, we can use the linearized equations or the adjoint state equations. In the computations that we present below, we will assume that $F_C$ is given by (8) and the modifications for treating case (9) are straightforward. We

will denote by $\delta_{\mathbf{Q}} J(\mathbf{Q}_C, \mathbf{b}_C)(\delta\mathbf{Q})$ the directional derivative of $J$ with respect to $\mathbf{Q}_C$ in the direction $\delta\mathbf{Q}$ and by $\delta_{\mathbf{b}} J(\mathbf{Q}_C, \mathbf{b}_C)(\delta\mathbf{b})$ the directional derivative of $J$ with respect to $\mathbf{b}$ in the direction $\delta\mathbf{b}$. Next, we present the expressions for the previous directional derivatives considering the linearized equations and the adjoint state equations, considering the two approaches for the computation of Dirac delta.

- Directional derivative of $J$ with respect to $\mathbf{Q}$ using the linearized equations:

$$\delta_{\mathbf{Q}} J(\mathbf{Q}_C, \mathbf{b}_C)(\delta\mathbf{Q}) = 2 \sum_{i=1}^{N_p} \sum_{n=1}^{N} (c_A^n(\mathbf{x}_i) - \widetilde{c}_i^n) \delta_{\mathbf{Q}} c_A^n(\mathbf{x}_i), \tag{20}$$

where, given $\delta_{\mathbf{Q}} c_A^0 = 0$, $\delta_{\mathbf{Q}} c_A^n \in W_h^A$, $n = 0, \dots, N-1$, is the solution to:

$$\alpha \int_{\Omega_A^{3D}} \delta_{\mathbf{Q}} c_A^{n+1} z \, d\mathbf{x} + \int_{\Omega_A^{3D}} K_C \nabla \delta_{\mathbf{Q}} c_A^{n+1} \cdot \nabla z \, d\mathbf{x} - \int_{\Omega_A^{3D}} G(\theta_A^{n+1}) \delta_{\mathbf{Q}} c_A^{n+1} z \, d\mathbf{x}$$
$$= \delta Q^{n+1} \int_{\Omega_A^{3D}} \varphi_{\mathbf{b}_C, \epsilon} z \, d\mathbf{x} + \alpha \int_{\Omega_A^{3D}} (\delta_{\mathbf{Q}} c_A^n \circ X_h^n) z \, d\mathbf{x}, \quad \forall z \in W_h^A, \tag{21}$$

in case (19). In case (18) $\delta_{\mathbf{Q}} c_A^n \in W_h^A$ is the solution to:

$$\alpha \int_{\Omega_A^{3D}} \delta_{\mathbf{Q}} c_A^{n+1} z \, d\mathbf{x} + \int_{\Omega_A^{3D}} K_C \nabla \delta_{\mathbf{Q}} c_A^{n+1} \cdot \nabla z \, d\mathbf{x} - \int_{\Omega_A^{3D}} G(\theta_A^{n+1}) \delta_{\mathbf{Q}} c_A^{n+1} z \, d\mathbf{x}$$
$$= \delta Q^{n+1} z(\mathbf{b}_C) + \alpha \int_{\Omega_A^{3D}} (\delta_{\mathbf{Q}} c_A^n \circ X_h^n) z \, d\mathbf{x}, \quad \forall z \in W_h^A. \tag{22}$$

- Directional derivative of $J$ with respect to $\mathbf{b}$ using the linearized equations:

$$\delta_{\mathbf{b}} J(\mathbf{Q}_C, \mathbf{b}_C)(\delta\mathbf{b}) = 2 \sum_{i=1}^{N_p} \sum_{n=1}^{N} (c_A^n(\mathbf{x}_i) - \widetilde{c}_i^n) \delta_{\mathbf{b}} c_A^n(\mathbf{x}_i), \tag{23}$$

where, given $\delta_{\mathbf{b}} c_A^0 = 0$, $\delta_{\mathbf{b}} c_A^{n+1} \in W_h^A$, $n = 0, \dots, N-1$, is the solution to

$$\alpha \int_{\Omega_A^{3D}} \delta_{\mathbf{b}} c_A^{n+1} z \, d\mathbf{x} + \int_{\Omega_A^{3D}} K_C \nabla \delta_{\mathbf{b}} c_A^{n+1} \cdot \nabla z \, d\mathbf{x} - \int_{\Omega_A^{3D}} G(\theta_A^{n+1}) \delta_{\mathbf{b}} c_A^{n+1} z \, d\mathbf{x}$$
$$= Q_C^{n+1} \int_{\Omega_A^{3D}} \delta_{\mathbf{b}} \varphi_{\mathbf{b}, \epsilon}(\delta\mathbf{b}) z \, d\mathbf{x} + \alpha \int_{\Omega_A^{3D}} (\delta_{\mathbf{b}} c_A^n \circ X_h^n) z \, d\mathbf{x}, \quad \forall z \in W_h^A, \tag{24}$$

in case (19). In case (18) $\delta_{\mathbf{b}} c_A^{n+1} \in W_h^A$ is the solution to:

$$\alpha \int_{\Omega_A^{3D}} \delta_{\mathbf{b}} c_A^{n+1} z \, d\mathbf{x} + \int_{\Omega_A^{3D}} K_C \nabla \delta_{\mathbf{b}} c_A^{n+1} \cdot \nabla z \, d\mathbf{x} - \int_{\Omega_A^{3D}} G(\theta_A^{n+1}) \delta_{\mathbf{b}} c_A^{n+1} z \, d\mathbf{x}$$
$$= Q_C^{n+1} \nabla z(\mathbf{b}_C) \cdot \delta\mathbf{b} + \alpha \int_{\Omega_A^{3D}} (\delta_{\mathbf{b}} c_A^n \circ X_h^n) z \, d\mathbf{x}, \quad \forall z \in W_h^A. \tag{25}$$

**Remark 2.** *It should be noted that in Equation (25) the term $Q_C^{n+1} \nabla z(\mathbf{b}_C) \cdot \delta\mathbf{b}$ appears. This term is the result of the approximation of the derivative of the Dirac delta [24,25]. The basic idea is to consider a polynomial approximation $\delta_h(\cdot, \mathbf{b})$ of the Dirac delta $\delta(\mathbf{x} - \mathbf{b})$. Indeed, let us assume that $\mathbf{b} \in T \in \tau_h^A$ and let $\widehat{\phi}(\widehat{\mathbf{x}}) \in \mathcal{P}_1(\widehat{T})$ such that*

$$\int_{\widehat{T}} \widehat{p}(\widehat{\mathbf{x}}) \widehat{\phi}(\widehat{\mathbf{x}}) = \widehat{p}(F_T(\mathbf{b})), \quad \forall \widehat{p} \in \mathcal{P}_1(\widehat{T}), \tag{26}$$

*where $\widehat{T}$ is the reference element and $F_T : T \to \widehat{T}$, with $F_T(\mathbf{x}) \in [\mathcal{P}_1(T)]^3$. To obtain the above polynomial, let us consider a basis $\widehat{B} = \{\widehat{p}_1, \widehat{p}_2, \dots, \widehat{p}_L\}$ of the vector space $\mathcal{P}_1(\widehat{T})$. We know that (26) is equivalent to:*

$$\int_{\widehat{T}} \widehat{p}_k(\widehat{\mathbf{x}})\widehat{\phi}(\widehat{\mathbf{x}}) = \widehat{p}_k(F_T(\mathbf{b})), \quad \forall k = 1, \ldots, L.$$

*We denote by $\widehat{\phi}_B = (\widehat{\alpha}_1, \ldots, \widehat{\alpha}_L)^t$ the coordinates of $\widehat{\varphi}$ on the basis of $\widehat{B}$ ($\widehat{\phi} = \sum_{l=1}^L \widehat{\alpha}_l \widehat{p}_l$). We know that $\widehat{\phi}_B$ is the solution to the following linear system:*

$$G_{\widehat{B}}\widehat{\phi}_{\widehat{B}} = \widehat{p}_{\widehat{B}}(F_T(\mathbf{b})),$$

*where $\widehat{p}_{\widehat{B}} = (\widehat{p}_1, \ldots, \widehat{p}_L)^t$ and $G_{\widehat{B}} \in Sym(M_{L \times L}(\mathbb{R}))$ is the Gram matrix associated with $\widehat{B}$:*

$$[G_{\widehat{B}}]_{i,j} = \int_{\widehat{T}} \widehat{p}_i(\widehat{\mathbf{x}})\widehat{p}_j(\widehat{\mathbf{x}}) \, d\widehat{\mathbf{x}}, \quad i,j = 1, \ldots, L.$$

*Thus,*

$$\begin{aligned}
\widehat{\phi}(\widehat{\mathbf{x}}) &= \; < \widehat{\phi}_B, \widehat{p}_{\widehat{B}}(\widehat{\mathbf{x}}) > \\
&= \; < G_{\widehat{B}}^{-1}\widehat{p}_{\widehat{B}}(F_T(\mathbf{b})), \widehat{p}_{\widehat{B}}(\widehat{\mathbf{x}}) > \\
&= \; < \widehat{p}_{\widehat{B}}(F_T(\mathbf{b})), G_{\widehat{B}}^{-1}\widehat{p}_{\widehat{B}}(\widehat{\mathbf{x}}) > .
\end{aligned} \tag{27}$$

*Now, we define*

$$\delta_h(\mathbf{x}, \mathbf{b}) = \begin{cases} |J_{\mathbf{x}}F_T(\mathbf{x})|\widehat{\phi}(F_T(\mathbf{x})) & if \quad \mathbf{x} \in T, \\ 0 & if \quad \mathbf{x} \notin T, \end{cases}$$

*where $J_{\mathbf{x}}F_T(\mathbf{x})$ is the Jacobian matrix of $F_T$. If we use expression (27):*

$$\delta_h(\mathbf{x}, \mathbf{b}) = \begin{cases} |J_{\mathbf{x}}F_T(\mathbf{x})| < \widehat{p}_{\widehat{B}}(F_T(\mathbf{b})), G_{\widehat{B}}^{-1}\widehat{p}_{\widehat{B}}(F_T(\mathbf{x})) > & if \quad \mathbf{x} \in T, \\ 0 & if \quad \mathbf{x} \notin T \end{cases}$$

*Taking into account the above definition, given an element $z_h \in W_h^A$:*

$$\begin{aligned}
\int_{\Omega_A^{3D}} \delta_h(\mathbf{x}, \mathbf{b})z_h(\mathbf{x}) \, d\mathbf{x} &= \int_T |J_{\mathbf{x}}F_T(\mathbf{x})| < \widehat{p}_{\widehat{B}}(F_T(\mathbf{b})), G_{\widehat{B}}^{-1}\widehat{p}_{\widehat{B}}(F_T(\mathbf{x})) > z_h(\mathbf{x}) \, d\mathbf{x} \\
&= \; < \widehat{p}_{\widehat{B}}(F_T(\mathbf{b})), G_{\widehat{B}}^{-1}\int_T |J_{\mathbf{x}}F_T(\mathbf{x})|\widehat{p}_{\widehat{B}}(F_T(\mathbf{x}))z_h(\mathbf{x}) \, d\mathbf{x} > \\
&= \; < \widehat{p}_{\widehat{B}}(F_T(\mathbf{b})), G_{\widehat{B}}^{-1}\int_{\widehat{T}} \widehat{p}_{\widehat{B}}(\widehat{\mathbf{x}})\widehat{z}_h(\widehat{\mathbf{x}}) \, d\widehat{\mathbf{x}} >,
\end{aligned}$$

*where $\widehat{z}_h(\widehat{\mathbf{x}}) = z_h(F_T^{-1}(\widehat{\mathbf{x}}))$. Now, $\widehat{z}_h = \widehat{z}_{h_B}^t \widehat{p}_{\widehat{B}}$; therefore:*

$$\begin{aligned}
< \widehat{p}_{\widehat{B}}(F_T(\mathbf{b})), G_{\widehat{B}}^{-1}\int_{\widehat{T}} \widehat{p}_{\widehat{B}}(\widehat{\mathbf{x}})\widehat{z}_h(\widehat{\mathbf{x}}) \, d\widehat{\mathbf{x}} > &= \; < \widehat{p}_{\widehat{B}}(F_T(\mathbf{b})), G_{\widehat{B}}^{-1}\int_{\widehat{T}} \widehat{p}_{\widehat{B}}(\widehat{\mathbf{x}})\widehat{z}_{h_B}^t\widehat{p}_{\widehat{B}}(\widehat{\mathbf{x}}) \, d\widehat{\mathbf{x}} > \\
&= \; < \widehat{p}_{\widehat{B}}(F_T(\mathbf{b})), G_{\widehat{B}}^{-1}G_{\widehat{B}}\widehat{z}_{h_B} > \\
&= \; \widehat{z}_{h_B}^t\widehat{p}_{\widehat{B}}(F_T(\mathbf{b})) \\
&= \; \widehat{z}_h(F_T(\mathbf{b})) \\
&= \; z_h(\mathbf{b}).
\end{aligned}$$

*Thus:*

$$\int_{\Omega_A^{3D}} \delta_h(\mathbf{x}, \mathbf{b})z_h(\mathbf{x}) \, d\mathbf{x} = z_h(\mathbf{b}).$$

*Finally:*

$$\int_{\Omega_A^{3D}} \nabla_{\mathbf{b}} \delta_h(\mathbf{x}, \mathbf{b}) z_h(\mathbf{x}) \, d\mathbf{x} = \nabla_{\mathbf{b}} < \widehat{p}_{\widehat{B}}(F_T(\mathbf{b})), G_{\widehat{B}}^{-1} G_{\widehat{B}} \widehat{z}_{h_B} >$$

$$= \widehat{z}_{h,B}^t J_{\widehat{\mathbf{x}}} \widehat{p}_{\widehat{B}}(F_T(\mathbf{b})) J_{\mathbf{x}} F_T(\mathbf{b})$$

$$= \nabla z_h(\mathbf{b}).$$

In view of the expressions (20) and (23), we observe that to calculate the gradient of the objective function using the linearized equations, we have to solve $N$ times (21) or (22) and 3 times (24) or (25). Indeed,

$$\frac{\partial J}{\partial Q^1}(\mathbf{Q}_C, \mathbf{b}_C) = \delta_{\mathbf{Q}} J(\mathbf{Q}_C, \mathbf{b}_C)(1, 0, \dots, 0),$$

$$\frac{\partial J}{\partial Q^2}(\mathbf{Q}_C, \mathbf{b}_C) = \delta_{\mathbf{Q}} J(\mathbf{Q}_C, \mathbf{b}_C)(0, 1, \dots, 0),$$

$$\vdots \qquad \vdots$$

$$\frac{\partial J}{\partial Q^N}(\mathbf{Q}_C, \mathbf{b}_C) = \delta_{\mathbf{Q}} J(\mathbf{Q}_C, \mathbf{b}_C)(0, 0, \dots, 1),$$

and

$$\frac{\partial J}{\partial b^1}(\mathbf{Q}_C, \mathbf{b}_C) = \delta_{\mathbf{b}} J(\mathbf{Q}_C, \mathbf{b}_C)(1, 0, 0),$$

$$\frac{\partial J}{\partial b^2}(\mathbf{Q}_C, \mathbf{b}_C) = \delta_{\mathbf{b}} J(\mathbf{Q}_C, \mathbf{b}_C)(0, 1, 0),$$

$$\frac{\partial J}{\partial b^3}(\mathbf{Q}_C, \mathbf{b}_C) = \delta_{\mathbf{b}} J(\mathbf{Q}_C, \mathbf{b}_C)(0, 0, 1).$$

Therefore, to calculate, for example, $\frac{\partial J}{\partial Q^1}(\mathbf{Q}_C, \mathbf{b}_C)$, we have to solve (21) or (22) taking $\delta_{\mathbf{Q}} = (1, 0, \dots, 0)$, for computing $\frac{\partial J}{\partial Q^2}(\mathbf{Q}_C, \mathbf{b}_C)$ and we have to solve (21) or (22) taking $\delta_{\mathbf{Q}} = (0, 1, \dots, 0)$, and so on.

Next, we will see that if we use the adjoint state equations it will only be necessary to solve one equation to calculate the gradient of the cost functional.

- Directional derivative of $J$ with respect to $\mathbf{Q}$ and $\mathbf{b}$ using the adjoint state equation. On the one hand,

$$\delta_{\mathbf{Q}, \mathbf{b}} J(\mathbf{Q}_C, \mathbf{b}_C)(\delta \mathbf{Q}, \delta \mathbf{b}_C) = 2 \sum_{i=1}^{N_p} \sum_{n=1}^{N} (c_A^n(\mathbf{x}_i) - \widetilde{c}_i^n) \delta_{\mathbf{Q}, \mathbf{b}} c_A^n(\mathbf{x}_i), \tag{28}$$

where, given $\delta c_A^0 = \delta_{\mathbf{Q}, \mathbf{b}} c_A^0 = 0$, $\delta c_A^{n+1} = \delta_{\mathbf{Q}, \mathbf{b}} c_A^{n+1} \in W_h^A$, $n = 0, \dots, N-1$ is the solution to:

$$\alpha \int_{\Omega_A^{3D}} \delta c_A^{n+1} z \, d\mathbf{x} + \int_{\Omega_A^{3D}} K_C \nabla \delta c_A^{n+1} \cdot \nabla z \, d\mathbf{x} - \int_{\Omega_A^{3D}} G(\theta_A^{n+1}) \delta c_A^{n+1} z \, d\mathbf{x}$$

$$= \delta Q^{n+1} \int_{\Omega_A^{3D}} \varphi_{\mathbf{b}_C, \epsilon} z \, d\mathbf{x} + Q_C^{n+1} \int_{\Omega_A^{3D}} \delta_{\mathbf{b}_C} \varphi_{\mathbf{b}, \epsilon}(\delta \mathbf{b}) z \, d\mathbf{x} \tag{29}$$

$$+ \alpha \int_{\Omega_A^{3D}} (\delta c_A^n \circ X_h^n) z \, d\mathbf{x}, \quad \forall z \in W_h^A,$$

in case (19). In case (18), $\delta c_A^{n+1} = \delta_{\mathbf{Q},\mathbf{b}} c_A^{n+1} \in W_h^A$ is such that:

$$
\alpha \int_{\Omega_A^{3D}} \delta c_A^{n+1} z \, d\mathbf{x} + \int_{\Omega_A^{3D}} K_C \nabla \delta c_A^{n+1} \cdot \nabla z \, d\mathbf{x} - \int_{\Omega_A^{3D}} G(\theta_A^{n+1}) \delta c_A^{n+1} z \, d\mathbf{x}
$$
$$
= \delta Q^{n+1} z(\mathbf{b}_C) + Q_C^{n+1} \nabla z(\mathbf{b}_C) \cdot \delta \mathbf{b} + \alpha \int_{\Omega_A^{3D}} (\delta c_A^n \circ X_h^n) z \, d\mathbf{x}, \quad \forall z \in W_h^A. \tag{30}
$$

Now, we will see how we can obtain the equations for the adjoint state. Let us consider $\{r_A^n\}_{n=0}^N \subset W_h^A$ such that $r_A^N = 0$ and let us take, for each $n = 1, \ldots, N-1$, $r_A^n$ as a test function in (29) or (30):

$$
\sum_{n=0}^{N-1} \left\{ \alpha \int_{\Omega_A^{3D}} \delta c_A^{n+1} r_A^n \, d\mathbf{x} + \int_{\Omega_A^{3D}} K_C \nabla c_A^{n+1} \cdot \nabla r_A^n \, d\mathbf{x} - \int_{\Omega_A^{3D}} G(\theta_A^{n+1}) \delta c_A^{n+1} r_A^n \, d\mathbf{x} \right\}
$$
$$
= \sum_{n=0}^{N-1} \left\{ \alpha \int_{\Omega_A^{3D}} (\delta c_A^n \circ X_h^n) r_A^n \, d\mathbf{x} + \delta Q^{n+1} \int_{\Omega_A^{3D}} \varphi_{\mathbf{b}_C, \epsilon} r_A^n \, d\mathbf{x} \right.
$$
$$
\left. + Q_C^{n+1} \int_{\Omega_A^{3D}} \delta_{\mathbf{b}_C} \varphi_{\mathbf{b}, \epsilon}(\delta \mathbf{b}) r_A^n \, d\mathbf{x} \right\},
$$

in case (19) and for (18):

$$
\sum_{n=0}^{N-1} \left\{ \alpha \int_{\Omega_A^{3D}} \delta c_A^{n+1} r_A^n \, d\mathbf{x} + \int_{\Omega_A^{3D}} K_C \nabla c_A^{n+1} \cdot \nabla r_A^n \, d\mathbf{x} - \int_{\Omega_A^{3D}} G(\theta_A^{n+1}) \delta c_A^{n+1} r_A^n \, d\mathbf{x} \right\}
$$
$$
= \sum_{n=0}^{N-1} \left\{ \alpha \int_{\Omega_A^{3D}} (\delta c_A^n \circ X_h^n) r_A^n \, d\mathbf{x} + \delta Q^{n+1} r_A^n(\mathbf{b}_C) + Q_C^{n+1} \nabla r_A^n(\mathbf{b}_C) \cdot \delta \mathbf{b} \right\}.
$$

Taking into account that $\delta c_A^0 = r_A^N = 0$,

$$
\sum_{n=0}^{N-1} \left\{ \alpha \int_{\Omega_A^{3D}} \delta c_A^{n+1} r_A^n \, d\mathbf{x} + \int_{\Omega_A^{3D}} K_C \nabla c_A^{n+1} \cdot \nabla r_A^n \, d\mathbf{x} - \int_{\Omega_A^{3D}} G(\theta_A^{n+1}) \delta c_A^{n+1} r_A^n \, d\mathbf{x} \right\}
$$
$$
= \sum_{n=0}^{N-1} \left\{ \alpha \int_{\Omega_A^{3D}} (\delta c_A^{n+1} \circ X_h^{n+1}) r_A^{n+1} \, d\mathbf{x} + \delta Q^{n+1} \int_{\Omega_A^{3D}} \varphi_{\mathbf{b}_C, \epsilon} r_A^n \, d\mathbf{x} \right. \tag{31}
$$
$$
\left. + Q_C^{n+1} \int_{\Omega_A^{3D}} \delta_{\mathbf{b}_C} \varphi_{\mathbf{b}, \epsilon}(\delta \mathbf{b}) r_A^n \, d\mathbf{x} \right\}
$$

in case (19) and for (18):

$$
\sum_{n=0}^{N-1} \left\{ \alpha \int_{\Omega_A^{3D}} \delta c_A^{n+1} r_A^n \, d\mathbf{x} + \int_{\Omega_A^{3D}} K_C \nabla c_A^{n+1} \cdot \nabla r_A^n \, d\mathbf{x} - \int_{\Omega_A^{3D}} G(\theta_A^{n+1}) \delta c_A^{n+1} r_A^n \, d\mathbf{x} \right\}
$$
$$
= \sum_{n=0}^{N-1} \left\{ \alpha \int_{\Omega_A^{3D}} (\delta c_A^{n+1} \circ X_h^{n+1}) r_A^{n+1} \, d\mathbf{x} + \delta Q^{n+1} r_A^n(\mathbf{b}_C) + Q_C^{n+1} \nabla r_A^n(\mathbf{b}_C) \cdot \delta \mathbf{b} \right\}. \tag{32}
$$

Therefore, if we define $r_A^n \in W_h^A$, $n = N-1, \ldots, 0$, as the solution to:

$$
\alpha \int_{\Omega_A^{3D}} r_A^n z \, d\mathbf{x} + \int_{\Omega_A^{3D}} K_C \nabla r_A^n \cdot \nabla z \, d\mathbf{x} - \int_{\Omega_A^{3D}} G(\theta_A^{n+1}) r_A^n z \, d\mathbf{x}
$$
$$
= \alpha \int_{\Omega_A^{3D}} (z \circ X_h^{n+1}) r_A^{n+1} \, d\mathbf{x} + 2 \sum_{i=1}^{N_P} (c_A^{n+1}(\mathbf{x}_i) - \widetilde{c}_i^{n+1}) z(\mathbf{x}_i), \tag{33}
$$

we know that:

$$\delta_{\mathbf{Q},\mathbf{b}} J(\mathbf{Q}_C, \mathbf{b}_C)(\delta\mathbf{Q}, \delta\mathbf{b}_C) \quad = \quad \sum_{n=0}^{N-1} \left\{ \delta Q^{n+1} \int_{\Omega_A^{3D}} \varphi_{\mathbf{b}_C,\epsilon} \, r_A^n \, d\mathbf{x} \right.$$
$$\left. + Q_C^{n+1} \int_{\Omega_A^{3D}} \delta_{\mathbf{b}_C} \varphi_{\mathbf{b},\epsilon}(\delta\mathbf{b}) r_A^n \, d\mathbf{x} \right\}, \tag{34}$$

in case we take an approximation of the Dirac delta (19) and if we consider (18):

$$\delta_{\mathbf{Q},\mathbf{b}} J(\mathbf{Q}_C, \mathbf{b}_C)(\delta\mathbf{Q}, \delta\mathbf{b}_C) = \sum_{n=0}^{N-1} \left\{ \delta Q^{n+1} r_A^n(\mathbf{b}_C) + Q_C^{n+1} \nabla r_A^n(\mathbf{b}_C) \cdot \delta\mathbf{b} \right\}. \tag{35}$$

It should be noted that the equation for the discrete adjoint state (33) is valid for any choice of approximation of the Dirac delta. The considered approximation of the Dirac delta appears in the expression of the gradient of cost functionals (34) or (35). Furthermore, to calculate the gradient of the cost functional it is only necessary to solve the equation for the adjoint state once. Indeed, given $\{r_A^n\}_{n=0}^N \subset W_h^A$, with $r_A^N = 0$, the solution to (33) is

$$\frac{\partial J}{\partial Q^k}(\mathbf{Q}_C, \mathbf{b}_C) = \int_{\Omega_A^{3D}} \varphi_{\mathbf{b}_C,\epsilon} \, r_A^{k-1} \, d\mathbf{x}, \ k = 1, \ldots, N, \tag{36a}$$

$$\frac{\partial J}{\partial b^k}(\mathbf{Q}_C, \mathbf{b}_C) = \sum_{n=0}^{N-1} Q_C^{n+1} \int_{\Omega_A^{3D}} \frac{\partial \varphi_{\mathbf{b},\epsilon}}{\partial b^k}(\mathbf{b}_C) \, r_A^n \, d\mathbf{x}, \ k = 1, 2, 3, \tag{36b}$$

in this case we consider (19); for (18):

$$\frac{\partial J}{\partial Q^k}(\mathbf{Q}_C, \mathbf{b}_C) = r_A^{k-1}(\mathbf{b}_C), \ k = 1, \ldots, N, \tag{37a}$$

$$\frac{\partial J}{\partial b^k}(\mathbf{Q}_C, \mathbf{b}_C) = \sum_{n=0}^{N-1} Q_C^{n+1} \frac{\partial r_A^n}{\partial x_i}(\mathbf{b}_C), \ k = 1, 2, 3. \tag{37b}$$

## 3. Results and Discussion

In this section, we will present the results that we obtained in the numerical simulations. All the numerical simulations were carried out with scientific software FreeFem++ [23] interfaced with IPOPT [20] on a 2019 MacBook Pro (2.5 GHz Intel Core i5 with four kernels).

We considered a scale three-dimensional mesh composed of nine buildings with heights of, respectively, 8, 5, 4, 5, 6, 8, 8, 5 and 4 m (back to front and left to right), with the geometrical configuration presented in Figure 2 (the depth of the soil considered is 3 m).

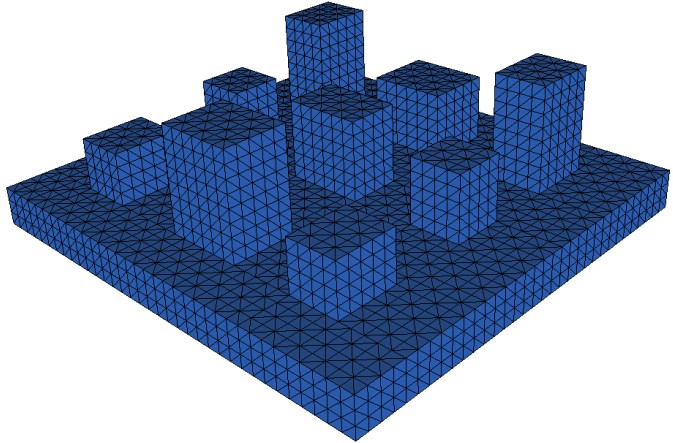

**Figure 2.** Geometrical configuration of the solid domain (soil and buildings, 6622 elements).

Associated with the previous geometrical configuration, we have considered the domain occupied by the air (effective computational domain for the control problem); in our case, we considered the upper boundary 10 m from the ground, see Figure 3.

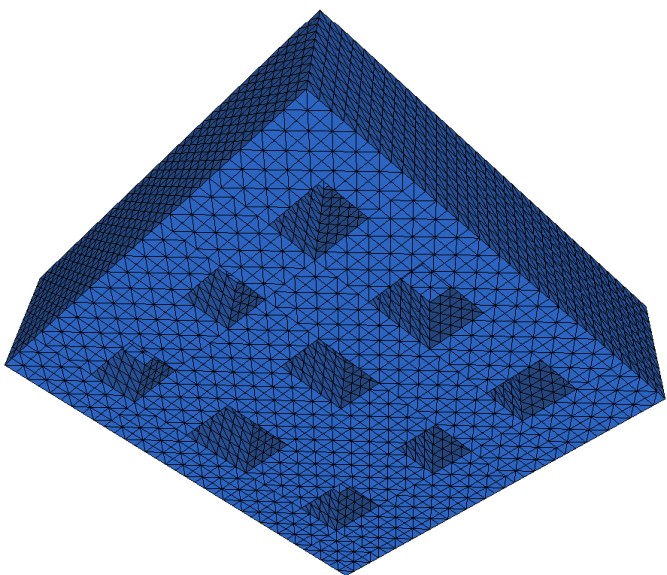

**Figure 3.** Computational domain for the control problem (12,736 elements).

In order to obtain the solution of the microclimate model (see Appendix A), we have considered the parameters listed in Table 1.

**Table 1.** Microclimate model parameters.

| Coefficient | Air | Asphalt | Buildings |
|---|---|---|---|
| Density | $\rho_A = 1.16 \times 10^3$ | $\rho_C = 2.11 \times 10^6$ | $\rho_B = 2.3 \times 10^6$ |
| Specific heat | $cp_A = 1.007$ | $cp_C = 0.92$ | $cp_B = 0.88$ |
| Conductivity | $\alpha_A = 0.05$ | $\alpha_C = 0.06$ | $\alpha_B = 0.06$ |
| Emissivity | | $\epsilon_C = 0.95$ | $\epsilon_B = 0.95$ |
| Albedo | | $a_C = 0.05$ | $a_B = 0.08$ |

The convective heat transfer coefficients for the corresponding interfaces were $h_C^A = 100$ on $\Gamma_C^A$, $h_G^A = 100$ on $\Gamma_G^A$, $h_C^B = 100$ on $\Gamma_C^B$, $h_W^A = 100$ on $\Gamma_B^W$, and $h_R^A = 1$ on $\Gamma_B^R$.

To compute the radiation temperatures appearing in the heat equations for soil and buildings, we assumed that $R_{sw,net}(\mathbf{x}, t) = (RM_{sw,dir} + RM_{sw,diff}) \, \sigma(\mathbf{x}, t)$, and $R_{lw,dow}(\mathbf{x}, t) = RM_{lw,dow} \, \sigma(\mathbf{x}, t)$, where, for our particular problem, we considered $RM_{sw,dir} = 650 \, \text{Wm}^{-2}$, $RM_{sw,diff} = 350 \, \text{Wm}^{-2}$, and $RM_{lw,dow} = 450 \, \text{Wm}^{-2}$. The function $\sigma(\mathbf{x}, t) \in [0, 1]$ models the attenuation of the previous maximum values, taking into account the movement of the sun: effect of shadows, night and day, and so on. In our simplified case, we have assumed that $\sigma(\mathbf{x}, t) = \max\{\sin(2\pi t / 86{,}400), 0\}$, $\forall (\mathbf{x}, t) \in (\Gamma_C^A \cup \Gamma_G^A \cup \Gamma_B^R) \times [0, T]$ (that is, over soil and roofs we considered no attenuation due to shadows and radiation only depending on time). Finally, we considered the initial values $\mathbf{u}_A^0 = (0, 0)$ m s$^{-1}$, $\theta_A^0 = \theta_S^0 = \theta_B^0 = 300 \, K$, and the boundary conditions $\mathbf{u}_A^{IN} = (0, 10^{-4}, 0)$ m s$^{-1}$, $\theta_A^{IN} = 300 \, K$.

We also considered that $G = 0$, $c_A^0 = 0 \, \text{gr m}^{-3}$ and $K_C = 10^{-1} \, \text{m}^2 \, \text{s}^{-1}$ in Equation (16). For the time discretization, we have considered a final time $T = 10{,}800$ s and $\Delta t = 900$ s ($N = 12$ time steps). In Figure 4, we can see the air velocity $\mathbf{u}_A$ and the air temperature $\theta_A$ on the boundary $\partial\Omega_A^{3D}$ at time $T = 10{,}800$ s.

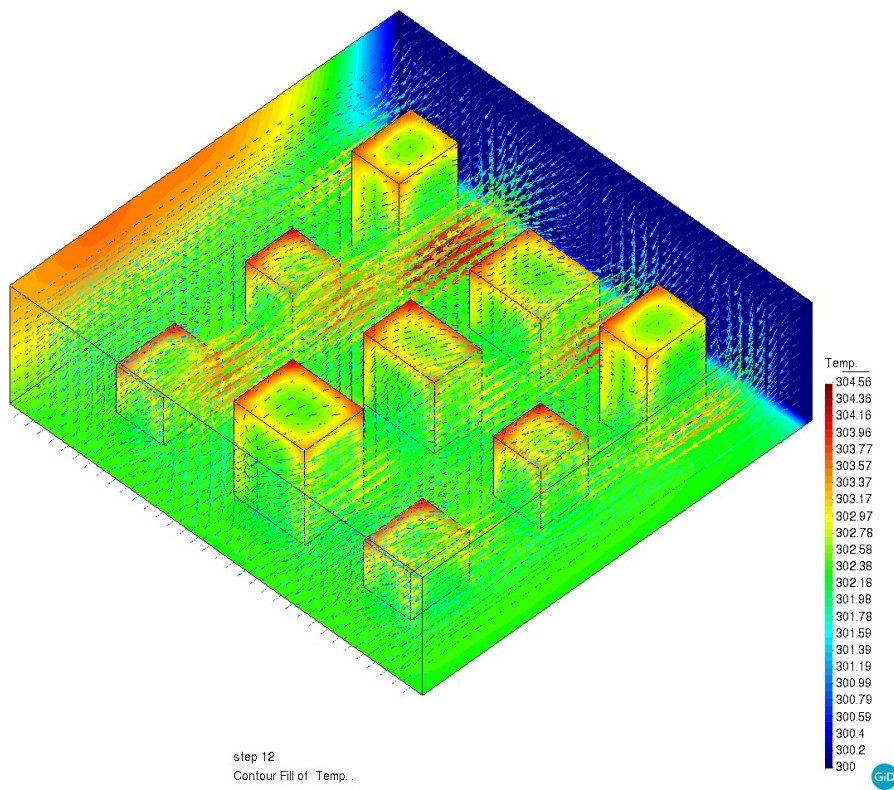

**Figure 4.** Microclimate model at time $T = 10{,}800$ s.

We will assume that in each building there are three sensors placed at 1, 2 and 3 m from the ground (see Figure 5). Therefore, we have a total of $N_p = 27$ measurement points.

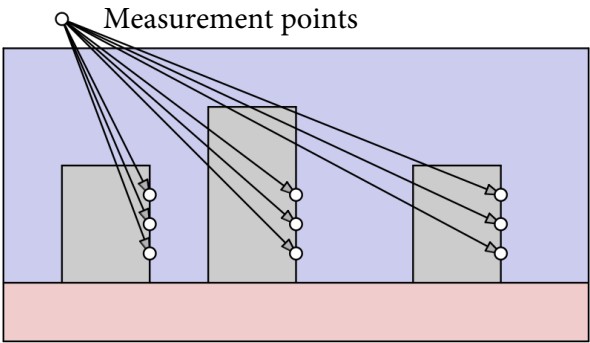

**Figure 5.** Measurement points in each building (1, 2 and 3 m from the ground).

We also assumed that we have measurements in each time step ($N_T = 12$). To validate the methodology proposed in this work, we have generated artificial measurements taking as the release point $\mathbf{b}_R = (2, 12, 6) \in Z_{ad}^2 = [1.2, 31.8] \times [10.2, 13.8] \times [4.2, 11.8]$ and $\mathbf{Q}_R = \{c_R(\mathbf{x}_i, t_n) : i = 1, \ldots, N_p, \ n = 1, \ldots, N_T\}$, where $c_R$ is the solution of the discretized state equation, Equation (17), associated with the release point $\mathbf{b}_R$ and the release rate $Q_R(t) = 3\left(1 + \sin(2\pi t / 86{,}400 - 2\pi \Delta t / 86{,}400)\right)$ gr m$^{-3}$ s$^{-1}$, $t \in [0, T]$, such that $Q_R(t) \in \mathcal{V}_{ad} = [0, 10]$. For instance, Figure 6 shows the concentrations in the measurement points placed at 3 m from the ground, if the delta approximation (19) is used. In this case, the distribution of the chemical agent at $T = 10{,}800$ s can be seen in Figure 7.

Therefore, the main objective of this section is to show how the methodology we propose allows one to recover the release point $\mathbf{b}_R$ and the discharge rate $\mathbf{Q}_R$ using artificial measures (see Figure 6).

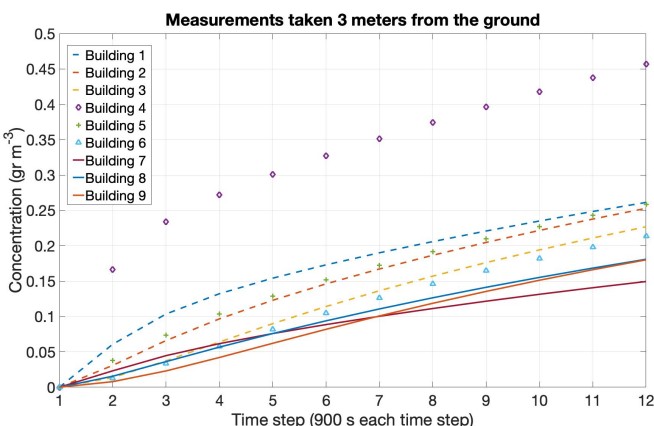

**Figure 6.** Concentration in the measurement points placed 3 m from the ground.

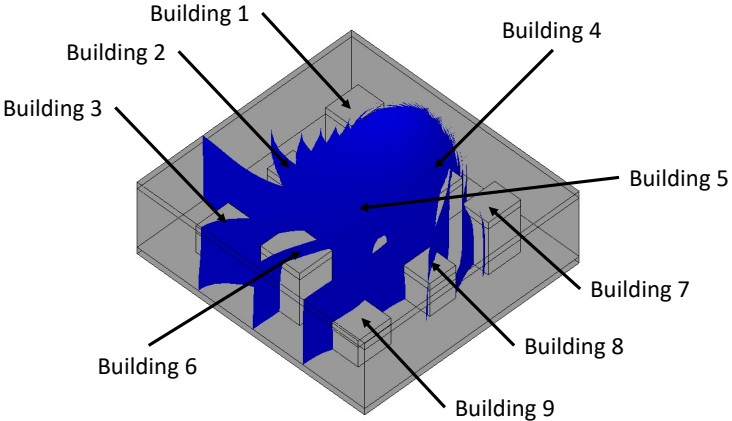

**Figure 7.** Chemical agent isosurfaces at $T = 10{,}800$ s.

We have obtained the following results taking a convergence tolerance for the IPOPT algorithm equal to $10^{-10}$:

1. **Problem 1, Equation** (10). **Estimation of the release point:** In this case, we fix the release rate to $\mathbf{Q}_R$ and we try to recover the release point $\mathbf{b}_R$ using the artificial measurements. To this end, we start from the initial control $\mathbf{b}_0 = [1.2, 10.2, 4.2] \in Z_{ad}^2$ and run the optimization algorithm in the following cases:

    (a) Delta approximation (19) and linearized Equation (23): 22 iterations, optimal cost $6.3054131 \times 10^{-22}$, CPU time 6502 s;

    (b) Delta approximation (19) and adjoint state Equation (36b): 32 iterations, optimal cost $3.2106931 \times 10^{-21}$, CPU time 4766 s;

    (c) Delta definition (18) and linearized Equation (23): 37 iterations, optimal cost $1.3769812 \times 10^{-20}$, CPU time 1779 s;

    (d) Delta definition (18) and adjoint state Equation (37b): 35 iterations, optimal cost $8.3016154 \times 10^{-21}$, CPU time 1958 s.

    In all the above cases, the release point used for the generation of artificial data is recovered by the algorithm.

2. **Problem 2, Equation** (11). **Estimation of the release rate:** In this case, we fix the release point to $\mathbf{b}_R$ and we try to recover the release rate $\mathbf{Q}_R$ using the artificial mea-

surements. We start from the initial control $\mathbf{Q}_0 = \mathbf{0} \in \mathcal{V}_{ad}$ and run the optimization algorithm in the following cases:

(a) Delta approximation (19) and linearized Equation (20): 26 iterations, optimal cost $1.8307661 \times 10^{-11}$, CPU time $11,177$ s.

(b) Delta approximation (19) and adjoint state Equation (36a): 29 iterations, optimal cost $1.8307668 \times 10^{-11}$, CPU time 4696 s.

(c) Delta definition (18) and linearized Equation (20): 25 iterations, optimal cost $1.8564142 \times 10^{-19}$, CPU time 2798 s.

(d) Delta definition (18) and adjoint state Equation (37a): 27 iterations, optimal cost $1.0237655 \times 10^{-19}$, CPU time 709 s.

In all the above cases, the release rate used for the generation of artificial data is recovered by the algorithm.

3. **Problem 3, Equation** (12). **Estimation of the release point and rate.** We try to recover the release point $\mathbf{b}_R$ and the release rate $\mathbf{Q}_R$ using the artificial measurements. We start from the initial control $\mathbf{b}_0 = [1.2, 10.2, 4.2] \in Z_{ad}^2$ and $\mathbf{Q}_0 = \mathbf{0} \in \mathcal{V}_{ad}$ and run the optimization algorithm in the following cases:

(a) Delta approximation (19) and linearized Equation (28): 252 iterations, optimal cost $1.2557307 \times 10^{-15}$, CPU time $180,895$ s.

(b) Delta approximation (19) and adjoint state Equations (36a) and (36b): 263 iterations, optimal cost $1.1373912 \times 10^{-18}$, CPU time $90,977$ s.

(c) Delta definition (18) and linearized Equation (28): 209 iterations, optimal cost $1.1029007 \times 10^{-12}$, CPU time $40,522$ s.

(d) Delta definition (18) and adjoint state Equations (37a) and (37b): 211 iterations, optimal cost $4.4262276 \times 10^{-14}$, CPU time $14,785$ s.

In all the above cases, the release rate and point used for the generation of artificial data are recovered by the algorithm.

## 4. Conclusions

In this paper, we propose a methodology for source identification of chemical incidents in urban areas. To achieve this, the classic advection-diffusion equation was completed with a reaction term depending on the air temperature, combined with a 3D microclimatic model, and numerically solved within the framework of mixed integer nonlinear programming (MINLP). In view of the results obtained, we observe that the proposed methodology is effective in identifying a source of contamination produced by a chemical agent in the academic cases studied. Both the proposed Dirac delta approximation and its own definition provide us with comparable results. The numerical resolution of the optimization problem using the adjoint state equations is more effective from the point of view of CPU time. Combining the search for the release point and rate requires a very high number of algorithm iterations, which penalizes CPU time.

**Author Contributions:** Writing, review and editing, F.J.F. and M.E.V.-M. All authors contributed equally. All authors have read and agreed to the published version of the manuscript.

**Funding:** This work has been partially supported by the Xunta de Galicia grant ED431C 2019/02 for Competitive Reference Research Groups (2019–2022).

**Data Availability Statement:** The authors confirm that the data supporting the findings of this study are available within the article.

**Conflicts of Interest:** The authors declare no conflict of interest. The funders had no role in the design of the study; in the collection, analyses, or interpretation of data; in the writing of the manuscript, or in the decision to publish the results'.

**Appendix A**

In this appendix, we include the microclimate mathematical model that we have used in the numerical simulations. The microclimate model is similar to that used by authors in [14]. We summarize it here for convenience of the reader. So, we consider a 2D domain $\Omega^{2D} = \{(x,y) \in \mathbb{R}^2 : 0 < x < a, \ 0 < y < b\}$ and two positive functions $H_S, H_A : \overline{\Omega^{2D}} \to [0, \infty)$ that represent the height of the layers of soil and air. We also consider a subdomain $\Omega_B^{2D} \subset \Omega^{2D}$ corresponding to the buildings and a function $H_B : \overline{\Omega_B^{2D}} \to [0, \infty)$ representing the height of these buildings. Then, we define the following 3D domains:

$$
\begin{aligned}
\Omega_S^{3D} &= \{(x,y,z) \in \mathbb{R}^3 : (x,y) \in \Omega^{2D}, \ 0 < z < H_S(x,y)\}, \\
\Omega_B^{3D} &= \{(x,y,z) \in \mathbb{R}^3 : (x,y) \in \Omega_B^{2D} \ H_S(x,y) < z < H_S(x,y) + H_B(x,y)\}, \\
\Omega_A^{3D} &= \{(x,y,z) \in \mathbb{R}^3 : (x,y) \in \Omega^{2D}, \ H_S(x,y) < z < H_S(x,y) + H_A(x,y)\} \setminus \Omega_B^{3D},
\end{aligned}
$$

which correspond, respectively, to the domain occupied by soil, buildings and the air. We can see that $\partial\Omega_S^{3D} = \Gamma_0^S \cup \Gamma_S^A \cup \Gamma_S^B \cup \Gamma_S^N$, $\partial\Omega_A^{3D} = \Gamma_S^A \cup \Gamma_B^R \cup \Gamma_B^W \cup \Gamma_A^H \cup \Gamma_A^{IN} \cup \Gamma_A^{OUT} \cup \Gamma_A^N$ and $\partial\Omega_B^{3D} = \Gamma_S^B \cup \Gamma_B^R \cup \Gamma_B^W$, where:

- $\Gamma_0^S$ is the lower boundary of the soil;
- $\Gamma_S^A$ is the interface boundary between air and soil;
- $\Gamma_S^B$ is the interface boundary between soil and buildings;
- $\Gamma_B^R$ is the boundary of buildings associated with roofs;
- $\Gamma_B^W$ is the boundary of buildings associated with walls;
- $\Gamma_A^H$ is the upper boundary of the air;
- $\Gamma_A^{IN}$ is the wind inflow boundary;
- $\Gamma_A^{OUT}$ is wind outflow boundary;
- $\Gamma_A^N$ is the lateral boundaries for the air;
- $\Gamma_S^N$ is the lateral boundaries for the soil.

We consider the following equations that model the behavior of the air temperature $\theta_A$ $(K)$, density $\rho_A$ $(\mathrm{m}^2 \, \mathrm{s}^{-2})$ and velocity $\mathbf{u}_A$ $(\mathrm{m} \, \mathrm{s}^{-1})$:

$$
\begin{cases}
\dfrac{\partial \mathbf{u}_A}{\partial t} + \nabla \mathbf{u}_A \mathbf{u}_A - \nabla \cdot (\nu_A \nabla \mathbf{u}_A) + \nabla p_A = \beta \theta_A \mathbf{g}, \ \Omega_A^{3D} \times ]0, T[, \\[2mm]
\nabla \cdot \mathbf{u}_A = 0, \ \Omega_A^{3D} \times ]0, T[ \\[2mm]
\mathbf{u}_A = u_A^{IN}, \ \Gamma_A^{IN} \times ]0, T[, \\[2mm]
(\nu_A \nabla \mathbf{u}_A + p_A I) \cdot \mathbf{n}_A - \dfrac{1}{2}(\mathbf{u}_A \cdot \mathbf{n}_A)_- \mathbf{u}_A = 0, \ \Gamma_A^{OUT} \times ]0, T[, \\[2mm]
\mathbf{u}_A = 0, \ \partial\Omega_A^{3D} \setminus (\Gamma_A^{IN} \cup \Gamma_A^{OUT}) \times ]0, T[, \\[2mm]
\mathbf{u}_A(0) = \mathbf{u}_A^0, \ \Omega_A^{3D},
\end{cases}
\tag{A1}
$$

where $\nu_A$ is the kinematic viscosity coefficient, $\theta_A^{REF}$ is a reference temperature, $\mathbf{g}$ is the gravity acceleration, $\mathbf{n}_A$ is the unit outward normal vector to the boundary $\partial\Omega_A^{3D}$, and $u_A^{IN}$, $u_A^{OUT}$ and $\mathbf{u}_A^0$ are given boundary and initial conditions. Observe that we are considering a directional do-nothing condition for the Navier–Stokes equations [26] on boundary $\Gamma_A^{OUT}$.

$$\begin{cases} \dfrac{\partial \theta_A}{\partial t} + \mathbf{u}_A \cdot \nabla \theta_A - \nabla \cdot (K_A \nabla \theta_A) = F_A, \ \Omega_A^{3D} \times ]0, T[, \\[2mm] \theta_A = \theta_A^{IN}, \ \Gamma_A^{IN} \times ]0, T[, \\[2mm] K_A \dfrac{\partial \theta_A}{\partial \mathbf{n}_A} = 0, \ (\Gamma_A^N \cup \Gamma_A^{OUT} \cup \Gamma_A^H) \times ]0, T[, \\[2mm] K_A \dfrac{\partial \theta_A}{\partial \mathbf{n}_A} = b_1^{S,A}(\theta_S - \theta_A), \ \Gamma_S^A \times ]0, T[, \\[2mm] K_A \dfrac{\partial \theta_A}{\partial \mathbf{n}_A} = b_1^{W,A}(\theta_B - \theta_A), \ \Gamma_B^W \times ]0, T[, \\[2mm] K_A \dfrac{\partial \theta_A}{\partial \mathbf{n}_A} = b_1^{R,A}(\theta_B - \theta_A), \ \Gamma_B^R \times ]0, T[, \\[2mm] \theta_A(0) = \theta_A^0, \ \Omega_A^{3D}, \end{cases} \tag{A2}$$

where $K_A$ is the diffusion coefficient, $b_1^{S,A}$, $b_1^{W,A}$ and $b_1^{R,A}$ are the convection coefficients, $F_A$ is a heat source term, and $\theta_A^{IN}$ and $\theta_A^0$ are given boundary and initial conditions.

The soil temperature $\theta_S$ $(K)$:

$$\begin{cases} \dfrac{\partial \theta_S}{\partial t} - \nabla \cdot (K_S \nabla \theta_S) = F_S, \ \Omega_S^{3D} \times ]0, T[, \\[2mm] K_S \dfrac{\partial \theta_S}{\partial \mathbf{n}_S} = b_1^{A,S}(\theta_A - \theta_S) + b_2^{A,S}((T_r^{A,S})^4 - \theta_S^4), \ \Gamma_S^A \times ]0, T[, \\[2mm] K_S \dfrac{\partial \theta_S}{\partial \mathbf{n}_S} = b_1^{B,S}(\theta_B - \theta_S), \ \Gamma_S^B \times ]0, T[, \\[2mm] K_S \dfrac{\partial \theta_S}{\partial \mathbf{n}_S} = 0, \ \Gamma_S^N \times ]0, T[, \\[2mm] \theta_S = \theta_S^{SUB}, \ \Gamma_S^0 \times ]0, T[, \\[2mm] \theta_S(0) = \theta_S^0, \ \Omega_S^{3D}, \end{cases} \tag{A3}$$

where $K_S$ is the diffusion coefficient, $b_1^{A,S}$ and $b_1^{B,S}$ are the convection coefficients, $b_2^{A,S}$ is the radiation coefficient, $T_r^{A,S}$ is the radiation temperature induced by solar radiation, $F_S$ is a source term, and $\theta_S^{SUB}$ and $\theta_S^0$ are given boundary and initial conditions.

The buildings temperature $\theta_S$ $(K)$:

$$\begin{cases} \dfrac{\partial \theta_B}{\partial t} - \nabla \cdot (K_B \nabla \theta_B) = F_B, \ \Omega_B^{3D} \times ]0, T[, \\[2mm] K_B \dfrac{\partial \theta_B}{\partial \mathbf{n}_B} = b_1^{A,R}(\theta_A - \theta_B) + b_2^{A,R}((T_r^{A,R})^4 - \theta_B^4), \ \Gamma_B^R \times ]0, T[, \\[2mm] K_B \dfrac{\partial \theta_B}{\partial \mathbf{n}_B} = b_1^{A,W}(\theta_A - \theta_B) + b_2^{A,W}((T_r^{A,W})^4 - \theta_B^4), \ \Gamma_B^W \times ]0, T[, \\[2mm] K_B \dfrac{\partial \theta_B}{\partial \mathbf{n}_B} = b_1^{S,B}(\theta_S - \theta_B), \ \Gamma_S^B \times ]0, T[, \\[2mm] \theta_S(0) = \theta_S^0, \ \Omega_S^{3D}, \end{cases} \tag{A4}$$

where $K_B$ is the diffusion coefficient, $b_1^{A,R}$, $b_1^{A,W}$ and $b_1^{S,B}$ are the convection coefficients, $b_2^{A,R}$ and $b_2^{A,W}$ are the radiation coefficients, $T_r^{A,R}$ and $T_r^{A,W}$ are the radiation temperatures, $F_B$ is a source term, and $\theta_B^0$ is a given initial condition.

The characteristic parameters that define the thermal behavior of the materials involved in the problem are the following:

- $\rho_A$, $\rho_S$ and $\rho_B$ (g m$^{-3}$) are the densities of air, soil and buildings, respectively;

- $cp_A$, $cp_S$ and $cp_B$ (Ws g$^{-1}$ K$^{-1}$) are the specific heat capacities of air, soil and buildings;
- $\alpha_A$, $\alpha_S$ and $\alpha_B$ (Ws g$^{-1}$ K$^{-1}$) are the thermal conductivities of air, soil and buildings;
- $\epsilon_S$, $\epsilon_W$ and $\epsilon_R$ (dimensionless constants) are the emissivities of the surfaces corresponding to soil, walls and roofs, respectivel;
- $a_S$, $a_W$ and $a_R$ (dimensionless constants) are the albedos of soil, walls and roofs, representing the ratio of reflected radiation from the surface to incident radiation upon it;
- $h_S^A$, $h_S^B$, $h_W^A$ and $h_R^A$ (W m$^{-2}$ K$^{-1}$) are the convective heat transfer coefficients between soil/air, soil/buildings, walls/air and roofs/air, respectively.

From the above coefficients, we can define the coefficients associated with Equations (A1)–(A4):

- $K_A$, $K_S$ and $K_B$ (m$^2$ s$^{-1}$) are the thermal diffusivities of air, soil and buildings, defined from the above data in the following way:

$$K_A = \frac{\alpha_A}{\rho_A\, cp_A}, \quad K_S = \frac{\alpha_S}{\rho_S\, cp_S}, \quad K_B = \frac{\alpha_B}{\rho_B\, cp_B}.$$

- $b_1^{S,A}$, $b_1^{A,S}$, $b_1^{S,B}$, $b_1^{B,S}$, $b_1^{W,A}$, $b_1^{A,W}$, $b_1^{R,A}$ and $b_1^{A,R}$ (m s$^{-1}$) are the coefficients related to convective heat transfer, obtained from the following relations:

  – for the temperature of air:

  $$\rho_A cp_A b_1^{S,A} = h_S^A, \quad \rho_A cp_A b_1^{W,A} = h_W^A, \quad \rho_A cp_A b_1^{R,A} = h_R^A.$$

  – for the temperature of soil:

  $$\rho_S cp_S b_1^{A,S} = h_S^A, \quad \rho_S cp_S b_1^{B,S} = h_S^B.$$

  – for the temperature of buildings:

  $$\rho_B cp_B b_1^{S,B} = h_S^B, \quad \rho_B cp_B b_1^{A,W} = h_W^A, \quad \rho_B cp_B b_1^{A,R} = h_R^A.$$

- $b_2^{A,S}$, $b_2^{A,W}$ and $b_2^{A,R}$ (m s$^{-1}$ K$^{-3}$) are the coefficients related to radiative heat transfer for soil, walls and roofs, respectively, obtained from following relations:

$$\rho_S cp_S b_2^{A,S} = \sigma_B \epsilon_S, \quad \rho_B cp_B b_2^{A,W} = \sigma_B \epsilon_W, \quad \rho_B cp_B b_2^{A,R} = \sigma_B \epsilon_R,$$

with $\sigma_B$ (W m$^{-2}$ K$^{-4}$) the Stefan–Boltzmann constant.

- Finally, in order to compute the radiation temperatures $T_r^{A,S}$, $T_r^{A,W}$ and $T_r^{A,R}$ (K) on the different solid boundaries (soil, walls and roofs), we use the following expressions (involving the corresponding solar radiations, albedos and emissivities):

$$\sigma_B \epsilon_S (T_r^{A,S})^4 = (1 - a_S) R_{sw,net}(\mathbf{x}, t) + R_{lw,dow}(\mathbf{x}, t),$$
$$\sigma_B \epsilon_W (T_r^{A,W})^4 = (1 - a_W) R_{sw,net}(\mathbf{x}, t) + R_{lw,dow}(\mathbf{x}, t),$$
$$\sigma_B \epsilon_R (T_r^{A,R})^4 = (1 - a_R) R_{sw,net}(\mathbf{x}, t) + R_{lw,dow}(\mathbf{x}, t),$$

where $R_{sw,net}(\mathbf{x}, t)$ denotes the net incident shortwave radiation on the surface, and $R_{lw,dow}(\mathbf{x}, t)$ denotes the downwelling longwave radiation, both measured in W m$^{-2}$.

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
