# Peer review of "Source Identification of a Chemical Incident in an Urban Area"

_axioms, doi:10.3390/axioms10030177_

Round 1

Reviewer 1 Report

I have read the manuscript very carefully, I really enjoyed seeing that diffusion equations can have such useful applications in real life. I think the manuscript has the necessary merit to be published.

Reviewer 2 Report

In this paper, the authors investigated an inverse source problem for a parabolic type equation from additional observation. On the basis of Optimal control theory, the authors proposed an algorithm for its numerical resolution. Finally several numerical examples were given to show the efficiency of the proposed methods.   In my opinion, some of results are interesting for some specialists. However, the manuscript is not well prepared and some places are confusing. I recommend a major revision. The attachment includes the mistakes and problems in the manuscript which need to be further addressed.

Reviewer 3 Report

The presented work is correctly structured, the discussed issues belong to the thematic area of the journal.

The considered mathematical model was presented with all the expected details. Additionally, Appendix A is a valuable and interesting part of the article.

The designed numerical algorithms are of an application nature. The obtained simulation results were correctly presented.

The disadvantage of this work is the lack of its clear positioning among the results of other researchers dealing with similar issues. 

There is one citation of numerous works that seems to be rather forced  than justified (line 117, 13-14, 15-18).

The literature on this subject is very rich, but it was not quoted by the authors.

In my opinion, the article should be placed in the context of modern scientific knowledge.

Reviewer 4 Report

The paper deals with the simulation of a chemical incident in an urban area and how to identify the source. The work is clearly written and well motivated. The results represent a considerable improvement upon present approaches. I can recommend its publication in Axioms. Only minor changes need to be addressed, which are listed below.

Minor points:

(1) Figure 2,3 are not quoted in the text. Please do that in the standard way.

(2) Neither Table 1 is quoted. Please modify the text in order to bring it into   the standard way.

(3) Line 188: write the tolerance in exponential format.
    The same holds in lines 194, 196, 198 and 200. And,
    lines 208, 210, 212, 214. Also, lines 222, 224, 226, 228.

(4) Explain why Div.V_A=0 is assumed (continuity equation for air motion?) above Eq.(8).

(5) Explain the meaning of p1 and p2 in the definition of Z_ad^k (Sect. 2.2).

(6) Below Eq.(12). I would write the following sentence in the form:
    'Let us observe that problems 1 and 3, Eqs.(10) and (12), respectively,
     can be formulated...'
(6a) Similarly below, please refer to Problem 3 (Eq.12), and Problem 1 (Eq.10).

(7) As a general remark, you should refer to Eq.(NN) instead of to
    Problem (NN). Please adapt the text in the way that the reader
    can clearly understand the meaning. Specially in lines 102-107.
    For instance, in line 189, there appears Problem (10), which is
    ok, but the distinction between problem and equation must be 
    addressed.

Reviewer 5 Report

The authors presented a method and algorithm for identifying the diffusion of chemicals in cities based on Mixed-Integer Nonlinear Programming.

Several suggestions for improving the manuscript.

1. The scientific novelty of the research is not clearly presented. I suggest detailed indicating the novelty of the proposed method and numerical algorithm both in the annotation and in the conclusion.

2. I suggest supplementing the literature review with a reference to the previous authors' research   http://dx.doi.org/10.1016/j.cam.2014.10.023 and show the development of the method used.

3. It is obvious that the height of buildings has a significant impact on the parameters of the gas diffusion process. I suggest clarifying the height of buildings in a numerical example and designate these heights in Figure 5 and especially in Figure 6.

4. The units of time and concentration measurement should be shown on the axes of the graph in Figure 6.

Round 2

Reviewer 2 Report

The current manuscript is a major revision to its original submission. The authors carried out point-to-point modifications according to the suggestions and comments by the referee. In comparison with the previous version, the overall scientific quality of the present manuscript is greatly improved, so that I am mostly satisfied with it. I recommend the publication